# Mesenchymal-Stem-Cell-Based Strategies for Retinal Diseases

**DOI:** 10.3390/genes13101901

**Published:** 2022-10-19

**Authors:** Xiteng Chen, Yuanfeng Jiang, Yanan Duan, Xiaomin Zhang, Xiaorong Li

**Affiliations:** Tianjin Key Laboratory of Retinal Functions and Diseases, Tianjin Branch of National Clinical Research Center for Ocular Disease, Eye Institute and School of Optometry, Tianjin Medical University Eye Hospital, Tianjin 300384, China

**Keywords:** mesenchymal stem cell, retinal diseases, cell therapy, exosome

## Abstract

Retinal diseases are major causes of irreversible vision loss and blindness. Despite extensive research into their pathophysiology and etiology, pharmacotherapy effectiveness and surgical outcomes remain poor. Based largely on numerous preclinical studies, administration of mesenchymal stem cells (MSCs) as a therapeutic strategy for retinal diseases holds great promise, and various approaches have been applied to the therapies. However, hindered by the retinal barriers, the initial vision for the stem cell replacement strategy fails to achieve the anticipated effect and has now been questioned. Accumulating evidence now suggests that the paracrine effect may play a dominant role in MSC-based treatment, and MSC-derived extracellular vesicles emerge as a novel compelling alternative for cell-free therapy. This review summarizes the therapeutic potential and current strategies of this fascinating class of cells in retinal degeneration and other retinal dysfunctions.

## 1. Introduction

The dysfunction or degeneration of retina, such as age-related macular degeneration (AMD), diabetic retinopathy (DR), and other hereditary diseases, are leading causes of irreversible blindness and vision loss which seriously impact the patient’s daily life [1]. Recently, the advances in surgical and medical management, such as intravitreal anti-VEGF (vascular endothelial growth factor) therapy, provide more options for retinal diseases [2,3], but are still invalid in many conditions especially when the disease is accompanied with retinal cell loss.

With the rapid development of stem cell research, stem-cell-based therapy is becoming one of the most promising strategies for untreatable retinal diseases. At present, plenty of stem cell clinical trials have been formally approved or are in progress. The clinical application of stem cell therapy will become an important landmark in the development of human medical history and radically change the whole picture of modern medicine [4,5,6,7]. Embryonic stem cells (ESCs), induced pluripotent stem cells (iPSCs), mesenchymal stem cells (MSCs), and many other adult stem cells are most commonly used in the laboratory studies and clinical trials, among which the rationales are not exactly the same. In some studies, the impaired retinal cells can be replaced by similar retinal cells derived from ESCs or iPSCs, while in other trials, neuroprotection and the paracrine effect of MSCs are dominant. MSCs may stand out as the best candidate for future clinical application, benefiting from their multiple functions, easy isolation, and low immunogenicity [8,9]. The outcomes of clinical trials in progress to test the potential of MSC-based therapy for retinal diseases are shown in Table 1.

MSCs are adult stem cells derived from embryonic mesoderm with a great capacity for self-renewal while maintaining their multipotency. Today, MSCs can be obtained from a wide variety of tissues including dental tissues, skin and foreskin, bones, muscles, salivary glands, blood, cartilages, and umbilical cords. Generally, MSCs can be isolated by two methods, the explant method and the enzymatic method. The key to the former approach lies in cutting the tissue into small pieces, in order to promote the diffusion of gases and nutrients into the cells. The latter method contains the additional steps of digesting the extracellular matrix with enzymes [21]. Former research believed that MSCs have multilineage potential, and they can home to the injured site and provide multiple benefits in wound healing, immune suppression, anti-inflammation, and neurotrophy. MSCs exhibit very low immunogenicity, and usually the patients need no systemic immunosuppressive medications when undergoing an autologous graft [22]. Numerous studies have demonstrated that human MSCs avoid allorecognition, interfere with dendritic cell and T-cell function, and generate a local immunosuppressive microenvironment by secreting cytokines [23]. In addition, MSCs are easy to be genetically manipulated, with a high metabolic activity and low intrinsic mutation rate, and can effectively secrete a variety of proteins [24]. At present, preclinical studies and clinical trials of MSC-based therapy have been performed in a variety of diseases with encouraging results, including autoimmune disease, vascular disease, organ transplantation, degenerative disease, nerve injury, joint reconstruction, and severe infection. In this article, we review the rationale for exploring MSCs as therapy for retinal dysfunction and discuss current and future advances in this area of research.

## 2. Retina Is a Suitable Target for MSC-Based Therapy

The retina is a continuation of the central nervous system, which is composed of neurons, glial cells, retinal pigment epithelial cells (RPE), and vasculature. Retinal diseases are involved with inflammation, immunity, trauma, neurodegeneration, angiopathy, and other pathological conditions. The special anatomy of ocular, diverse pathogenesis and remodeling progress of the retinal diseases make retina a more appropriate target for MSC-based therapy than other tissues. The advantages include the following: (i) the transparent refractive media allow us to observe the retina intuitively, while a variety of detection methods can provide accurate and vivid evaluation of retinal function and evaluate the process of therapy in vivo, as well as enable immediate response to the possible adverse effect accordingly; (ii) stem cells are more likely to survive inside the ocular environment, which is relatively immune privileged due to the blood–eye barrier [25]; (iii) retinal stem cell transplantation is feasible to be operated using the available clinical techniques of ophthalmology; (iv) plenty of repeatable animal models of retinal diseases ensure the smooth conduct and convincing of study (Figure 1).

Due to particular organizational structure of retina and the characteristics of retinal diseases, MSC-based treatment primarily concerns how to sufficiently suppress the inflammatory response, reduce tissue damage, and ameliorate blood supply and nutrition in order to promote tissue remodeling and neurological functions. Thus, appropriate administration routes of MSCs and strategies are required for the diversity of retinal diseases.

## 3. The Delivery Options

Generally, there are two administration routes of MSCs: local and systemic intravenous application (Table 2).

For the local administration route, MSCs are usually delivered intravitreally or subretinally by a microscope. Subretinal transplantation of MSCs can anatomically circumvent the blood-retinal barrier to the adjacent area of the outer retina and avoid the challenge of immune rejection. One of the commonly used methods in clinical trials is to inject stem cells directly through the retina after pars plana vitrectomy [26]. In some preclinical studies, researchers successfully implanted the cells through the sclera and choroid into the subretinal space [27]. Without passing into the vitreous, the incidence rate of proliferative vitreoretinopathy is reduced because of less interference on the intraocular environment. However, this method demands a master hand in order to avoid penetrating the retina into the vitreous, and moderate cell suspension liquid should be prepared as not to induce retinal detachment (RD). Long-term safety and retention of MSCs from humans in the subretinal space has been shown in Royal College of Surgeons rats with some preservation of retinal degeneration on histology [28].

An alternative approach to local stem cell therapy is intravitreal injection, an administration route that has been globally applied on account of the prevalence of anti-VEGF therapy. Unlike vitrectomy surgery for subretinal implantation of cells, which requires hospital admission and significant recovery time from surgery, intravitreal injection of cells can be operated in the clinic, with minimal damage and risk to the eye. It seems that intravitreal administration is more suitable for treatment of inner retinal lesions taking account of the anatomic location. Nevertheless, stem cells do not readily incorporate into the retina, partly resulting from the inherent integrity of the internal limiting membrane [29].

Intravenous injection can be repetitively performed at the clinic and barely causes any harm to the recipient. After systemic administration, MSCs can possibly reach the maximum range of optic circulation and benefit the entire retina and choroid [30,31]. MSCs can be recruited to injured retina following intravenous transplantation and adhere to vascular endothelium by adhesion behavior via vascular cell adhesion molecules [32]. Nevertheless, systemic injection requires a large number of cells and may increase the possibility of immune rejection. In addition, it is difficult for MSCs to migrate into the retina after intravenous transplantation, mostly due to blocking by the blood-retinal barrier.

However, with the deepening of MSC-based research, more researchers argued the theory of direct cell replacement, suggesting the revived and regenerative effect of MSCs by introducing its paracrine trophic effect [33,34]. If MSC-based therapy can play a paracrine role effectively by intravitreal or intravenous delivery without serious adverse effects, the ease of such routes of administration would be highly desired in the future. Such an MSC-based treatment would not be disease-specific and can have broad clinical applications.

## 4. Strategies for Treating the Retinal Diseases with Mesenchymal Stem Cells

As we know, there are encouraging outcomes in MSC-based treatments for retinal diseases (Figure 2).

### 4.1. Implications of Mesenchymal Stem Cell Therapy in Retinal Degeneration

Stem cell therapy has opened a new era of tissue regeneration. MSCs may provide an alternative source for retinal regeneration. In addition to the established multipotential of MSCs to differentiate into osteoblasts, adipocytes, and chondrocytes, they were seen to differentiate into retinal cells in early studies. Bone-marrow-derived MSCs (BMSCs) and adipose-tissue-derived MSCs (AMSCs) are shown to express RPE and photoreceptor cell markers by exposure to conditioned medium in vitro [35,36]. Intravenous injection of MSCs into diabetic rats resulted in the targeted migration of the transplanted MSCs to the damaged retina and the expression of astrocytes and photoreceptor cell markers. However, from a dialectical point of view, we should be aware that ectopic expression of some photoreceptor markers does not signify that cells differentiate into photoreceptors and does not mean these cells are functional. Researchers tried to replace damaged retinal cells by directional differentiation of MSCs in order to reconstruct the retinal neural network and restore the activity of nerve conduction [37,38]. In animal models of retinal laser injury, almost all MSCs migrated and integrated into the layers of the retina by intravitreal MSC injection [31,39]. Recently, Ripolles-Garcia A et al. showed that human donor photoreceptor precursor cells were generated in vitro from hESCs and were integrated and differentiated into the canine retina. Notably, these generated photoreceptors have synaptic pedicle-like structures which could establish contact with second-order neurons. Thus, their findings set the stage for evaluating functional vision restoration following photoreceptor replacement in canine models of inherited retinal degeneration [40].

In contrast, intravitreal administration of MSCs remained in the vitreous cavity and did not differentiate into neural or perivascular-like cells while preventing retinal ganglion cell loss by triggering an effective cytoprotective microenvironment in the models of diabetic and experimental ocular hypertension [41,42]. However, undifferentiated MSCs are able to express many mature neuronal and glial markers, such as βIII-tubulin and GFAP, which make them more controversial for early discoveries [43,44]. Conversely, they might be overly simplistic and optimistic for the concept of direct cell replacement, since retinal degeneration diseases usually involve several layers and cell types of retina. Instead, administrated MSCs with paracrine trophic effects and some capable of direct engraftment might allow the stem cells to maximize their potential in retinal regeneration [45,46,47,48].

In terms of long-term application, a study reported that 82 participants with retinitis pigmentosa (RP) successfully received 5 million UC-MSCs to the suprachoroidal area. After 6 months, studies showed improvement in best-corrected visual acuity (BCVA), visual field (VF), and multifocal electroretinography (mf-ERG) measurements in most patients, and no ocular or systemic adverse events were observed during the follow-up period [11]. Additionally, intravitreal injection of BM-MSCs also appears to be safe and effective for patients with RP. The procedures were performed safely and only one patient needed surgery for a serious adverse effect during the 12-month period [14]. Regardless of the genetic mutations in RP patients, subtenon transplantation of WJ-MSCs is also effective and safe in the first year of treatment, preventing the progression of the disease without any adverse effect [12]. In addition, suprachoroidal implantation of AD-MSCs also had a beneficial effect on vision, VF, and mfERG in patients with AMD and Stargardt’s disease in a 6-month follow-up study [49]. These studies confirmed the potential feasibility and security of MSC-based implantation in degenerative retinal diseases.

### 4.2. Immunosuppressive Effect of Mesenchymal Stem Cells in Uveitis

Chronic and recurrent uveitis involving the choroid and retina are usually treated by corticosteroids and immunosuppressants with many serious potential side effects, including myelosuppression, tumor formation, and liver and kidney damage. By targeting different parts of the immune system, MSCs are supposed to inhibit the autoimmune response, allograft rejection, and graft-versus-host disease [50]. Theoretically, MSCs are superior to corticosteroids and immunosuppressants in that they also possess neurotrophic and antimicrobial effects [51,52,53].

MSCs are currently being tested in preclinical studies for the treatment of experimental autoimmune uveitis (EAU) owing to their immunosuppressive properties [54]. Yang et al. reported four cases of refractory uveitis resistant to traditional systemic administration, which resolved after intravenous injection with human-umbilical-cord-derived MSCs (hUMSCs) [55]. A single course of MSC-based therapy at the onset of the disease suffices to protect against ocular inflammation in EAU [56]. Moreover, double treatments of MSCs with longer intervals have a significant curing effect [57]. Our studies indicated that MSC-based therapy is more effective in controlling inflammation, reducing relapses, and protecting the retina in recurrent EAU than dexamethasone. The beneficial effects of MSCs in recurrent EAU were attributed to a significant decrease in Th1/Th17-mediated inflammation, to regulation of the balance between Th17 and Tregs, and to suppression of the function of antigen-presenting cells [53,57]. Further study demonstrated that the immunomodulatory function of MSCs might be mediated through the CD73/adenosine pathway in human and mouse T cells. CD73 on MSCs, upregulated by transforming growth factor-β1 (TGF-β1), cooperated with CD39 and CD73 on activated T cells to produce adenosine, resulting in inhibition of T-cell proliferation [58]. Injection of MSCs regulated STAT1 and STAT6 phosphorylation to reduce the levels of migration-related proteins in dendritic cells (DCs) and inhibit the proliferation of DCs [59]. Recent research shows that MSCs continue to play a vital role in EAU, as they inhibit the activation of CD4+T cells by direct intercellular contact and activation of prostaglandin signaling [60].

### 4.3. Neuroprotective Effect of Mesenchymal Stem Cells in Reducing or Delaying the Retinal Tissue Damage

MSCs are able to regulate the toll-like receptor 4 signaling pathway and pro-inflammatory factors such as tumor necrosis factor-α, interleukin (IL)-1β, and reactive oxygen species to reduce retinal cell apoptosis, increase retinal inner layer thickness, and reduce neuroinflammation [61]. Furthermore, plenty of studies have shown that MSCs are capable of secreting a variety of neurotrophic factors, such as basic fibroblast growth factors (bFGF), ciliary neurotrophic factors, ganglion-cell-derived neurotrophic factors (GDNF), nerve growth factors (NGF), and brain-derived neurotrophic factors in the treatment of retinal diseases [62,63]. Under the effect of neurotrophic factors secreted by MSCs, conditioned medium of MSCs can promote the photoreceptor cell proliferation in vitro and inhibit their death [64].

In the animal model of retinal degeneration, rats suffering from retinitis pigmentosa were given subretinal injection or intravenous injection with MSCs, which successfully inhibits retinal degeneration progress and protects the function of the retina, but transplanted MSCs did not migrate or integrate into the retina [38,65,66]. As for animal models of retinal ischemia reperfusion injury and glaucoma, intravitreal injection of MSCs could significantly improve the retinal ganglion cell survival [42,67,68,69]. MSCs distributed along the inner limiting membrane expressed a variety of neurotrophic factors, but only a handful of stem cells could migrate into the retina. Coculture of MSCs with retinal explants also confirmed the reduction in apoptosis and increase in the nerve fiber layer and inner plexiform layer thicknesses. Cell secretome demonstrated that MSCs secrete a number of neuroprotective proteins and suggest that platelet-derived growth factor secretion in particular may play an important role in MSC-mediated retinal ganglion cell neuroprotection [70]. Our study also indicated that MSCs might display their therapeutic effect in a paracrine fashion by secreting neuroprotective and anti-inflammatory factors to preserve the homeostasis of the ECM and regulate the intraocular microenvironment, which is beneficial for the integrity of the retina and tissue repair [71]. This experimental evidence all points to the speculation that the paracrine effect of MSCs may play a key role in the method of cell protection on the retina instead of migrating into the retina or differentiating into retina cells. In these diseases, MSCs give a neurotrophic effect, mainly by secreting neurotrophic factors, in addition to other possible mechanisms of MSCs, which include regulating the inflammatory process, repairing the blood vessel damage, and promoting synaptic regeneration by adjusting the inhibiting signal to activate the intrinsic repair mechanisms.

Many clinical studies have shown that MSC-based surgery holds great potential in retinal diseases owing to its anti-inflammatory and neuroprotective effects. Özmert et al. observed that the subtenon Wharton’s jelly-derived MSCs transplant can be beneficial to patients with retinitis pigmentosa [12]. Injection of autologous BMSCs showed meaningful visual improvements in the vast majority of patients with nonarteritic ischemic optic neuropathy [19].

### 4.4. Tissue Repair and Inflammatory Chemotaxis of Mesenchymal Stem Cells

Pathological process of inflammatory reaction and tissue damage are involved in the above diseases such as uveitis, laser-induced retinal damage, ischemia injury, and diabetic retinopathy. MSCs play a role in anti-inflammation and promoting restoration of the retinal tissue damage. MSCs have the capacity to home into damaged tissue with inflammation, probably in response to chemokines, adhesion molecules, and matrix metalloproteinases, following intravenous injection [31,72]. However, the specific mechanism still remains a mystery [73]. In vitro results suggested that the stromal-derived factor 1/C-X-C chemokine receptor type 4 (SDF-1/CXCR4) and hepatocyte growth factor/c-met (HGF/c-met) axes, along with MMPs, act as important signals for migration and homing of mesenchymal stem cells [74,75,76,77]. An in vivo study also demonstrated that src family protein kinases are activated by SDF-1/CXCR4 signaling and play an essential role in SDF-1/CXCR4-mediated MSCs’ chemotactic response and ischemic cardiac recruitment [78].

### 4.5. Antioxidative Properties of Mesenchymal Stem Cells

ROS levels increase dramatically under environmental stress, which causes serious damage to cellular structures in retinal disease. As hMSCs were able to scavenge free radicals, promote endogenous antioxidant defenses, alter mitochondrial bioenergetics, transfer mitochondria to impaired cells, and effectively regulate oxidative stress, they have been studied as a treatment for oxidative injury. Ohkouchi et al. showed that MSCs elevated the survival rate of A549 cells through Stanniocalcin 1 upregulation [79]. Injection of AMSCs into the subretinal space of mice under oxidative stress protected and rescued RPE and photosensor cells [80].

### 4.6. Angiogenic Potentials of Mesenchymal Stem Cells

Abnormal angiogenesis is one of the main reasons for many ocular diseases, including DR, retinopathy of prematurity, and AMD. As MSCs can secrete angiogenesis-related factors and proteins, such as VEGF, fibroblast growth factor, HGF, TGF-β1, and insulin-like growth factor 1, they can promote angiogenesis and repair retinal ischemic injury [81,82,83].

Studies have shown that adult stem cells [84], AMSCs [85], iPSCs [25], and autologous BMSCs [17] are promising treatment options for animal models of DR and its complications. Elevated blood glucose levels in patients with DR lead to increased levels of reactive oxygen species (ROS), damaged pericytes and endothelial cells, vascular degeneration, and formation of new vessels. MSCs can not only differentiate into pericytes and reverse the changes in extracellular matrix proteins [86], but also inhibit the inflammatory response caused by oxidative stress through reducing the levels of pro-inflammatory factors, calcium, and ROS [87]. At the same time, the expression of intraocular nerve growth factor, bFGF, GDNF, and other nutritional factors, such as NGF and NT-3, can be induced to reduce nerve cell apoptosis [88].

In addition, MSCs also have anti-angiogenic effects in proliferative retinopathy. High levels of TGF-β1 secreted from human placental amniotic membrane derived MSCs demonstrated rescue potential on suppressing retinal neovascularization in a mouse model of oxygen-induced retinopathy [82].

### 4.7. Mitochondria Donation

Mitochondria create chemical energy for biochemical activities. A majority of mitochondrial proteins are encoded from nucleus DNA, maintaining mitochondrial functions, while mitochondria also contains its own DNA, known as mitochondrial DNA, which encodes 13 proteins [89]. Dysfunction of mitochondria is a signal indicating cellular senescence, and mitochondrial injury finally results in RPE cell death and degenerative retinal disease [90]. Both aging and hyperglycemia can lead to oxidative stress, damaging mitochondria and accelerating AMD and RD development [91]. In glaucoma, mitochondrial dysfunction is accompanied with retinal ganglion cell (RGC) degeneration [92], which is difficult to repair.

A fascinating and creative way that MSCs rescue impaired neural cells has provoked profound thought. Through different approaches, MSCs can deliver their own mitochondria to injured cells, in order to promote their repair and regeneration. This phenomenon was first described by Spees et al., who cocultured mitochondrial gene-depleted cells with MSCs. The mutant cells with enhanced mitochondria showed expression of mitochondrial proteins, and significantly increased ATP production and decreased lactate levels, a byproduct of anerobic respiration [93]. Previous studies have revealed several ways that mitochondria can be transferred, such as via tunneling nanotubes, gap junctions, or exosomes [94]. This kind of donation pathway is proven in ocular cells, including the corneal endothelium, RPE, and photoreceptors [95]. Intravitreal iPSC-derived MSC transplantation can significantly transfer mitochondria to damaged RGCs and improve retinal function [96]. Furthermore, Kim et al. overexpressed pigment epithelium derived factor, an antioxidative factor, in MSCs and cocultured it with oxidative-stress-impaired RPE. The outcome indicated enhancement of biogenesis regulators including NRF1, PPARGC1A, and TFAM, which are necessary for promoting mitochondrial transcription and respiration [97,98]. Such a unique and inspiring assistance mechanism specifies the potential advantages for future stem cell therapeutic strategy.

### 4.8. Restraint of Cell Migration—Retinal Barriers

The adult visual pathway can possibly rebuild new synaptic interactions and guide new axons in certain circumstances, which provide an opportunity for cellular therapy. However, being similar to the central nervous system, the retina is not easily influenced by the outside factors, for example, stem cells, due to the retinal barriers. The blood–retinal barrier is part of the blood–ocular barrier that consists of retinal vascular endothelium and RPE [99]. The physiological barrier of retinal blood vessels comprises a single layer of non-fenestrated endothelial cells which maintain the inner blood–retinal barrier. The tight junctions between retinal epithelial cells, which form the outer blood–retinal barrier, prevent the passage of large molecules from choriocapillaris into the retina. In addition, internal and external limiting membrane, extracellular matrix components (such as chondroitin sulfate proteoglycans), active RGC synapses, glial scars produced by reactive gliosis in the injury, and pathological conditions are also important parts of the retina barriers [100].

Whether using local or systemic administration, MSCs were not seen to have migrated or integrated into the retina on an ideal scale while the retinal barriers were relatively integrated [38,67,68]. Thus, these retinal barriers remain the major obstacle for directional differentiation of stem cells to replace damaged retinal cells. A previous study delineated that retinal MSCs’ migration correlated positively with the amount of peeled internal limiting membrane, and targeted disruption of glial reactivity, with α-aminoadipic acid treatment, dramatically improved the structural integration of intravitreally transplanted cells [101]. By the manipulation of mechanical injury, incorporation of grafted stem cells was seen in 50% of the injured retinas, as well as subsequent differentiation into the neuronal and glial lineage, and formation of synapse-like structures were implied in the adult rat retina [102]. Another study showed that matrix metalloproteinase-2 can promote host–donor integration by degrading CD44 and neurocan at the outer surface of the degenerative retina without disruption of the host retinal architecture [103]. In addition, subretinal injection with chondroitinase ABC combined with enhanced immune suppression caused a dramatic increase in the migration of stem cells into all the retinal cell layers [100].

MSCs also have limitations for human retinal disease treatment due to the vulnerability of their expression. Some new technologies, such as gene therapy, retinal organoids in vitro, and bio-printing technology, have demonstrated prospective therapeutic capabilities to repair damaged retinal cells.

Erythropoietin-engineered human MSCs enhance differentiation into retinal photoreceptors in retinal degenerative diseases [104]. HiPSC-derived RGCs are seeded on a biodegradable poly (lactic-co-glycolic acid) scaffold to create an engineered biomaterial [105]. Retinal-ganglion-like cells differentiated from dental pulp stem cells using 3D networks to replace the lost and damaged RGCs [106].

### 4.9. New Insights about Exosomes Derived from Mesenchymal Stem Cells

Exosomes are the tiniest extracellular vesicles with bi-lipid membranes shuttling active cargoes (for example genetic material, proteins, and other biologically active molecules) involved in the complex intercellular communication system [107]. Being released by various cell types (such as B and T cells, DCs, cancer cells, stem cells, and endothelial cells), the main traits of exosomes are in accordance with the function of their original cells.

Recent discoveries noticed that MSC-derived exosomes, 50–150 nm microvesicles, could inherit the multiple functions from MSCs and might be the key mediators of MSC paracrine activity [108,109]. They have been studied in various disease models with many encouraging results. In cardiovascular disease, the infarct size and cardiac function were ameliorated in myocardial ischemia/reperfusion injury and acute myocardial infarction following administration of exosomes derived from MSCs [110]. MSC-derived exosomes also play a role in inflammation regulation and ischemia/reperfusion-induced renal injury repair, partially by suppressing the recruitment and activation of macrophages related to the C-C motif chemokine receptor-2 expression [111]. Extracellular vesicles exhibit immunomodulatory properties similar to their original MSCs through inhibitory activity on B-cell proliferation, intervention of shifting T cells from an activated status to a T regulatory phenotype, reduction in interferon-γ production, and increased release of immunosuppressive cytokines (such as prostaglandin E2, TGF-β, IL-10, and IL-6) [112,113,114,115]. In addition, the protective effect of MSCs on acute lung injury can be potentiated by ischemic preconditioning through the secretion of exosomes [116]. In neurological diseases, a recent study demonstrated that systemic administration of cell-free exosomes generated by MSCs improves functional recovery in rats that suffered from traumatic brain injury, while another group proposed that combined MSCs and MSC-derived exosome therapy displayed the best result in reducing the brain infarct volume and preserving neurological function in rats after acute ischemic stroke [117,118].

Although preclinical studies of exosomes derived from stem cells have been applied to the treatment for a variety of diseases, very few results came out in ophthalmology. We first observed the therapeutic effect of exosomes derived from MSCs in retinal dysfunction [119]. MSC-derived exosomes were able to pass retinal barriers and diffuse throughout the retina after intravitreal injection. After laser injury, MSC/exosome-treated groups showed smaller lesioned areas, less TUNEL-positive cells, and better ERG responses. Further in vivo and in vitro experiments suggested that their suppression of injury-induced inflammation may be via the down-regulation of monocyte chemotactic protein-1.

While extant obstacles limit the clinical applications of simple stem cell treatment for retinal diseases, such as retinal barriers, the alternative application of MSC-derived exosomes remains more promising, primarily due to their inherited abilities, low immunogenicity, long half-life in circulation, and other cell-free advantages. Additionally, MSCs can produce a higher number of exosomes compared to other cells. However, whether MSC-derived exosomes preserve the same effective properties compared to the cells themselves remains controversial, as MSCs may act by releasing other active soluble factors and the biological features of exosomes may vary with different extracellular environments. Therefore, further studies are required to understand the full dimension of this complex intercellular communication system and how it can be optimized to take full advantage of its therapeutic effects.

## 5. The Pitfalls of Mesenchymal-Stem-Cell-Based Therapies

Despite many clinical advances in mesenchymal stem cell therapy in the past decades (Table 1), there are still many challenges. The main challenges include the immunocompatibility, stability, heterogeneity, differentiation, and migratory capacity.

Cell survival is the key to successful MSC-based therapy. After implantation, ischemia and the inflammatory microenvironment are likely to cause apoptosis of MSCs in vivo. In addition, both enzymatic treatment to separate MSCs from culture plates prior to transplantation and injection procedures are able to destroy cells and reduce their survival rate.

Moreover, the application of autologous MSCs is limited by time-consuming preparation and loss of cell viability. To overcome these limitations, allogeneic MSC transplantation has been applied. As it is easy to cause immune rejection, studies have shown that allogeneic BMSCs are easily eliminated [120]. In addition, studies have shown an increased risk of infection after transplantation. For example, allogeneic MSCs have only a transient response when treating hematopoietic-stem-cell-induced graft-versus-host disease (GVHD), and invasive fungal and viral infections have been reported in patients in some clinical trials [121]. It was suggested that this may be due to the inhibitory effect of BMSCs on thymic reconstitution and subsequent impairment of immune recognition. Therefore, close monitoring is required after MSC transplantation.

## 6. Discussion

Retinal dysfunctions, especially the ones that lead to retinal degeneration, are the leading causes of vision loss with very few effective treatments. The pleiotropic properties and therapeutic potentials of MSCs, including broad distribution, immune evasion, capacity for self-renewal and multilineage differentiation, and secretion of an array of anti-inflammatory, immunoregulatory, or neuroprotective mediators, bring new hope for the untreatable retinopathies. In general, the use of MSCs has been explored as a cellular therapy for retinal degenerative conditions through replacement of retinal cells and/or the release of protective factors into damaged tissue. However, in direct transplantation without in vitro induction, MSCs showed limited ability and extremely low efficiency for integration or differentiation toward retinal cells, as the main obstacle seems to be the integrity of retinal barriers. Even the ones that exhibited photoreceptor phenotypes lack convincing evidence of synaptic fusion into the damaged neuropil. Meanwhile, increasing lines of evidence point to the paracrine pattern as the key role of MSC-based strategies, displaying beneficial effects on retinal tissue remodeling and functional preservation via secreting cytokines, cell–cell interactions, and releasing extracellular vesicles. As an emerging research field, MSC-derived extracellular vesicles, especially exosomes, hold a great potential for cell-free therapies in retinal dysfunctions that are safer and easier to manipulate than the idea of cell replacement. However, further exploration and elucidation of the biologic features, therapeutic mechanisms, long-term safety and curative effects, and limitations and complications of MSC-based strategies are needed before the full potential can be unearthed.

## Figures and Tables

**Figure 1 genes-13-01901-f001:**
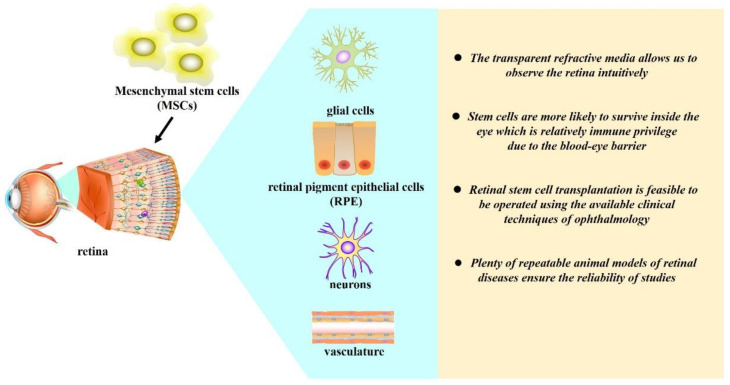
Retina is a suitable target for MSC-based therapy.

**Figure 2 genes-13-01901-f002:**
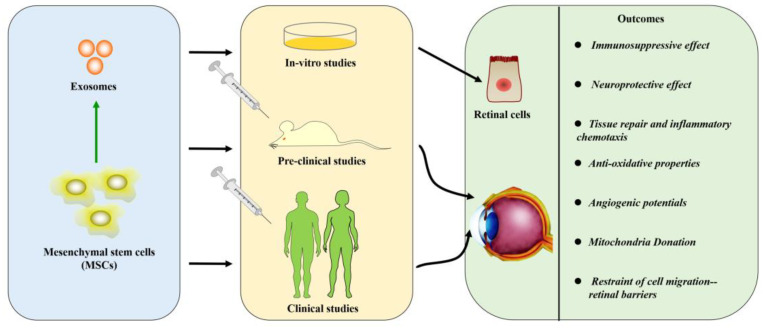
Potential effects and mechanisms for treating retinal diseases with mesenchymal stem cells.

**Table 1 genes-13-01901-t001:** MSC-based treatment in clinical trials for retinal disease.

Study (Year)	Retinal Disease	Number of Patients	Cell Type	Route of Administration	Dosage	Phase of Study	Outcomes
Oner [10] (2016)	RP	11	AD-MSC	Subretinal	2.47 × 10^6^ ± 0.11 cells/150 μl	I	Minor ocular complications
Kahraman [11] (2020)	RP	82	UC-MSC	Suprachoroidal	5 × 10^6^ cells/2 mL	III	Beneficial effect on BCVA, VF, and mfERG
Özmert [12] (2020)	RP	32	WJ-MSC	Subtenon	2–6 × 10^6^ cells/1.5 mL	III	No ocular or systemic adverse events
Weiss [13] (2018)	RP	17	BM-MSC	Retrobulbar, subtenon, intravitreal and intravenous	1.2 × 10^9^ cells/14–15 cm^3^	NA	Beneficial effect on visual acuity
Tuekprakhon [14] (2021)	RP	14	BM-MSC	Intravitreal	1 × 10^6^ cells, 5 × 10^6^ cells, 1 × 10^7^ cells	I	Beneficial effect on visual acuity
Park [15] (2014)	IDRD	6	BM-MSC	Intravitreal	-	I	No ocular or systemic adverse events associated with treatment
Siqueira [16] (2011)	HRD	6	BM-MSC	Intravitreal	10 × 10^6^ cells/0.1 mL	I	No ocular or systemic adverse events associated with treatment
Gu [17] (2018)	DR	17	BM-MSC	Intravenous	3 × 10^6^ cells/kg	NA	Beneficial effect on macular thickness and visual acuity
Levy [18] (2015)	Optic nerve diseases	1	BM-MSC	Retrobulbar, subtenon, intravitreal, and intravenous	1.2 × 10^9^ cells/14–15 cm^3^	NA	Beneficial effect on visual acuity
Weiss [19] (2017)	NAION	10	BM-MSC	Retrobulbar, subtenon, intravitreal, and intravenous	1.2 × 10^9^ cells/14–15 cm^3^	NA	Beneficial effect on visual acuity
Kuriyan [20] (2017)	AMD	3	AD-MSC	Intravitreal	-	NA	Severe ocular complications

RP: retinitis pigmentosa; BCVA: best-corrected visual acuity; VF: visual field; mfERG: multifocal electroretinography; IDRD: ischemic and degenerative retinal disorders; HRD: hereditary retinal dystrophy; DR: diabetic retinopathy; NAION: nonarteritic ischemic optic neuropathy; AMD: age-related macular degeneration; AD-MSC: adipose-tissue-derived mesenchymal stem cell; UC-MSC: umbilical-cord-derived mesenchymal stem cell; WJ-MSC: Wharton’s jelly-derived mesenchymal stem cell; BM-MSC: bone-marrow-derived mesenchymal stem cell.

**Table 2 genes-13-01901-t002:** The comparison of the delivery options in MSC-based treatment for retinal disease.

Local Administration	Systemic Administration
Intravitreal or subretinal injection	Intravenous injection
**Advantages**	
Anatomically circumventing the blood–retinal barrier and averting the challenge of immune rejection Having confirmed long-term safety and retention	Barely causing any harm to the patients MSCs can possibly reach the maximum range of optic circulation and benefit the entire retina and choroid
**Disadvantages**	
Demanding a master hand Potentially causing retinal detachment Requiring hospital admission and significant recovery time from surgery	Causing the possibility of immune rejection Mostly blocked by the blood–retinal barrier

## Data Availability

Not applicable.

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
