# Peer review of "Mesenchymal-Stem-Cell-Based Strategies for Retinal Diseases"

_genes, 2022, doi:10.3390/genes13101901_

Round 1

Reviewer 1 Report

Dear Authors,

Thanks very much for the manuscript.

Recently Cell and Gene Therapy are in the rise. It's fascinating to experience this era of modern medicine. This is, in particular, very significant in treating different life-threatening diseases, especially for those patients who didn’t respond to any other existing treatments.

At this time, the present manuscript on Stem Cell-based therapy for retinal Diseases is highly relevant.

The authors very nicely introduced the topic to the general and the specialized audience.

I would just like to give some comments to improve the manuscript; other than that, everything else looks good to me.

1.     Please make a table of currently available data of the top 5 clinical trials on this topic and show it in the introduction.

2.     Figure 1 of Advantages of…. Retina, the texts are not very easily understandable; you might consider changing it. Please mention some disadvantages.

3.     Please compare the delivery options in a tabular form.

4.     I would like the authors to recheck the recent research on this topic, e.g.

Ripolles-Garcia A et al. (https://www.cell.com/stem-cell-reports/fulltext/S2213-6711(22)00325-3) has recently shown that human donor photoreceptor precursor cells were generated in vitro from human embryonic stem cells were integrated and differentiated into the canine retina. This could be another strategy for treating blindness using mesenchymal stem cells.

5.     Texts in Figure 2 are also not easily understandable.

6.     Please add one outlook section (Discussion) where one can mention the pitfalls of stem cell-based therapies.

Thanking you,

Best regards,

Goutam

Reviewer 2 Report

I have several major comments to improve manuscript quality:

1/Globally, the authors should add a section describing the different sources of MSCs and the way they are prepared/isolated: BM, adipose…

1/ Citation 1 is restricted to 1 country : I would suggest other citations such as:

Blindness GBD, Vision Impairment C, Vision Loss Expert Group of the Global Burden of Disease S. 2021a. Causes of blindness and vision impairment in 2020 and trends over 30 years, and prevalence of avoidable blindness in relation to VISION 2020: the Right to Sight: an analysis for the Global Burden of Disease Study. Lancet Glob Health 9: e144-e160.

2/ Line 38 : The authors should specify the rational behind the different stem cell based strategies: either retinal cell replacement by similar retinal cells derived from hESC or hiPSCs or neuroprotection through MSCs or other paracrine acting cells.

3/ The manuscript should be revised by a native English speaker: some sentences are really difficult to read.

Line 43-45: missing word or rewrite the sentence

Line 58-59 : sentence not clear : the CNS is not composed of RPE.

4/ Line 111: the use of the words “stem cell” alone is not specific enough: it should be avoided through the manuscript: either use MSCs, hiPSCs or other more specific terms.

5/ Line 125: differentiation of MSCs into photoreceptors or RPE cells is controversial. Citing old papers that were never reproduced is not appropriate. Ectopic expression of some photoreceptor markers does not signify that cells differentiated into photoreceptors and does not means these cells are photosensitive (functional). Could they authors discuss these points.

6/ Line 240 ref 28: not related to ipscs

7/ Line 264 : “magical” : I understand the interest of authors but such word has no place in a scientific publication.

8/ Fig1: that’s not really the advantage of msc therapy but rather the advantage of targeting the eye with cell therapy

9/ Fig 2 does not present the strategies for treating retinal diseases. To be informative, it should present specific functions of MSCs in particular diseases, modes of delivery, if autologous or from an allogenic frozen cell bank,expected outcomes in particular functions,…

10/ Authors could discuss the long term impact of MSC on retinal degenerative diseases : if the effect is sustained or not (are they publications in this particular aspect ?) or discuss it with regard to the nature of deseases (gene deficiency for example in retinitis pigmentosa).

11/ As MSCs are evaluated in clinical trials, the authors should add a section describing ongoing clinical trials (perhaps a table) and discuss observed outcomes.

12/ The authors should draw attention to the troubles associated with the uncontrolled used of MSCs in “stem cell clinics” and the danger for patients.

Round 2

Reviewer 2 Report

The authors did not correctly addressed the following points:

10/ Authors could discuss the long term impact of MSC on retinal degenerative diseases : if the effect is sustained or not (are they publications in this particular aspect ?) or discuss it with regard to the nature of deseases (gene deficiency for example in retinitis pigmentosa).

Response: Thanks for the suggestion. We have added the long-term observation of MSC-based treatment for retinal degenerative diseases. Please see line 181-194.

Reviewer : The new paragraph does not discuss long term effect of MSCs ? but rather hESC based retinal cells !

12/ The authors should draw attention to the troubles associated with the uncontrolled used of MSCs in “stem cell clinics” and the danger for patients.

Re: Thanks for the suggestion, and we have added a section in the text. Please see line 422-440.

Reviewer : not really what I was expecting but ok.
